# FGF23 and Klotho Levels are Independently Associated with Diabetic Foot Syndrome in Type 2 Diabetes Mellitus

**DOI:** 10.3390/jcm8040448

**Published:** 2019-04-03

**Authors:** Javier Donate-Correa, Ernesto Martín-Núñez, Carla Ferri, Carolina Hernández-Carballo, Víctor G. Tagua, Alejandro Delgado-Molinos, Ángel López-Castillo, Sergio Rodríguez-Ramos, Purificación Cerro-López, Victoria Castro López-Tarruella, Miguel Angel Arévalo-González, Nayra Pérez-Delgado, Carmen Mora-Fernández, Juan F. Navarro-González

**Affiliations:** 1Research Unit, University Hospital Nuestra Señora de Candelaria (UHNSC), 38010 Santa Cruz de Tenerife, Spain; emarnu87@gmail.com (E.M.-N.); carlamferri@gmail.com (C.F.); carolinahdezcarballo@gmail.com (C.H.-C.); vtagua@funcanis.es (V.G.T.); carmenmora.fdez@gmail.com (C.M.-F.); 2Doctoral and Graduate School, University of La Laguna, 38200 San Cristóbal de La Laguna, Spain; 3Vascular Surgery Service, University Hospital Nuestra Señora de Candelaria, 38010 Santa Cruz de Tenerife, Spain; adelgadomolinos@gmail.com (A.D.-M.); angellopezcastillo24@gmail.com (Á.L.-C.); 4Transplant Coordination, University Hospital Nuestra Señora de Candelaria, 38010 Santa Cruz de Tenerife, Spain; sergiotomasr@hotmail.com (S.R.-R.); pcerlop@gobiernodecanarias.org (P.C.-L.); 5Pathology Service, University Hospital Nuestra Señora de Candelaria, 38010 Santa Cruz de Tenerife, Spain; vcaslop@gmail.com; 6Human Anatomy and Histology Department, University of Salamanca, 37008 Salamanca, Spain; marevalo@usal.es; 7Clinical Analysis Service, University Hospital Nuestra Señora de Candelaria, 38010 Santa Cruz de Tenerife, Spain; nperdel@gobiernodecanarias.org; 8Nephrology Service, University Hospital Nuestra Señora de Candelaria, 38010 Santa Cruz de Tenerife, Spain; 9Institute of Biomedical Technologies, University of La Laguna, 38200 San Cristóbal de La Laguna, Spain

**Keywords:** diabetic foot syndrome, vascular disease, fibroblast growth factor 23, Klotho, inflammation

## Abstract

Background: Diabetic foot syndrome (DFS) is a prevalent complication in the diabetic population and a major cause of hospitalizations. Diverse clinical studies have related alterations in the system formed by fibroblast growth factor (FGF)-23 and Klotho (KL) with vascular damage. In this proof-of-concept study, we hypothesize that the levels of FGF23 and Klotho are altered in DFS patients. Methods: Twenty patients with limb amputation due to DFS, 37 diabetic patients without DFS, and 12 non-diabetic cadaveric organ donors were included in the study. Serum FGF23/Klotho and inflammatory markers were measured by enzyme-linked immunosorbent assay (ELISA). Protein and gene expression levels in the vascular samples were determined by immunohistochemistry and quantitative real-time PCR, respectively. Results: Serum Klotho is significantly reduced and FGF23 is significantly increased in patients with DFS (*p* < 0.01). Vascular immunoreactivity and gene expression levels for Klotho were decreased in patients with DFS (*p* < 0.01). Soluble Klotho was inversely related to serum C-reactive protein (*r* = −0.30, *p* < 0.05). Vascular immunoreactivities for Klotho and IL6 showed an inverse association (*r* = −0.29, *p* < 0.04). Similarly, vascular gene expression of *KL* and *IL6* were inversely associated (*r* = −0.31, *p* < 0.05). Logistic regression analysis showed that higher Klotho serum concentrations and vascular gene expression levels were related to a lower risk of DFS, while higher serum FGF23 was associated with a higher risk for this complication. Conclusion: FGF23/Klotho system is associated with DFS, pointing to a new pathophysiological pathway involved in the development and progression of this complication.

## 1. Introduction

Type 2 diabetes mellitus (T2DM) has become a critical health problem, with more than 450 million people living with this disease worldwide [1]. Diabetic foot syndrome (DFS) is a prevalent complication in this population and a major cause of hospitalizations. This syndrome carries the risk of limb amputation, which represents the most prevalent non-traumatic amputation surgery in the hospital setting [2]. Mortality associated with DFS is similar to that of breast, prostate, or colon cancer [3]. Hence, there is an overwhelming need for a better understanding of the molecular mechanisms underlying the development of DFS and foremost to detect early those at the highest risk of this complication and develop more specific and effective therapies.

DFS has a complex and multifactorial pathogenesis where an underlying vascular atherogenic process affecting both the endothelium and the smooth muscle cell layer is a critical factor. Atherosclerosis is a chronic inflammatory phenomenon involving several pathways and molecules, including inflammatory cytokines [4,5]. Closely related with the atherogenic phenomena, the alterations of the mineral metabolism have gained special prominence as pathogenic factors in the development and progression of vascular damage [6,7].

The newly described system formed by the fibroblast growth factor (FGF)-23 and Klotho (KL) proteins is recognized as one of the main regulators of mineral metabolism. FGF23 is a phosphaturic hormone that is synthesized in the bone in response to dietary phosphate intake in order to keep a normal phosphate homeostasis [8]. To this end, it binds to cognate fibroblast growth factor receptors (FGFRs; primarily FGFR1 and 3) in the renal tubular cells in the presence of the obligated co-receptor Klotho, a type 1 trans-membrane protein predominantly expressed in the kidneys [8]. In addition, a soluble form of Klotho is generated by the shedding of the ectodomain by the A Disintegrin and A Metalloproteinase 17 (ADAM17), or by an alternative RNA splicing [9,10,11]. Beyond this physiological role, the FGF23/Klotho system has been related to various processes associated with cardiovascular damage.

Diverse clinical studies have directly related the increase in serum concentrations of FGF23 with cardiovascular damage and with an increased mortality [12,13]. Moreover, recent works suggest that high levels of FGF23 can exert deleterious effects on the cardiovascular system by establishing low affinity bonds with FGFRs, independently of Klotho [14,15]. On the other hand, reduced concentrations of soluble Klotho have been involved in processes related to premature aging, including the appearance of vascular damage [16]. This soluble form of Klotho is detectable in urine, serum, and cerebrospinal fluid, acting as a humoral factor with multiple functions such as anti-oxidation, modulation of ion channels, anti-Wnt signaling or anti-apoptosis, and anti-senescence effects [17]. Importantly, the expression of *KL* has been recently demonstrated in human vascular tissue [16,18] and both the levels of systemic and vascular Klotho have been related with cardiovascular complications [16,19,20,21].

In this proof-of-concept study, we hypothesize that the levels of FGF23 and Klotho are altered in DFS patients. We determined the serum concentrations and the vascular expression levels of FGF23 and Klotho in a group of diabetic patients, with and without DFS. We also measured the levels of inflammatory mediators, including the cytokines tumor necrosis factor (TNF)-α, interleukin (IL)-6 and IL10, and analyzed the relationship among these variables and the presence of DFS.

## 2. Materials and Methods

### 2.1. Patients

Eighty-five diabetic patients undergoing an elective open vascular surgery procedure due to established clinical atherosclerotic artery disease were considered for enrollment in this study. Exclusion criteria included hemodynamic instability; history of chronic inflammatory, immunologic, or tumoral disease; positive serology to hepatitis B, hepatitis C, or HIV; acute inflammatory or intercurrent infectious episodes in the previous month; institutionalization; treatment with immunotherapy or immunosuppressive drugs; previous organ transplantation and advanced renal disease (estimated glomerular filtration rate (eGFR) lower than 30 mL/min/1.73m^2^).

For comparative purposes in the immunohistochemical and gene expression analyses, femoral samples from 12 cadaveric non-diabetic organ donors matched by age, sex, and eGFR, and without any medical history of cardiovascular disease were recovered during organ retrieval surgery.

The study protocol was approved by the Institutional Ethics Committee of the University Hospital Nuestra Señora de Candelaria (Santa Cruz de Tenerife, Spain) and complied with ethical standards of the Declaration of Helsinki. Written informed consent was obtained from all participants before they participated in the study.

### 2.2. Samples and Biochemical Markers

During the surgical procedure, a sample of the carotid or femoral arteries, according to the affected vessel, was obtained from the participants. At the same time, serum samples were drawn, aliquoted, and immediately stored at −80 °C for further analysis. Routine biochemical parameters were determined using standard methods. Serum levels of intact FGF23 were measured by specific enzyme-linked immunosorbent assay (ELISA) (EMD Millipore Corporation, Milford, MA, USA), which detects only the intact form of FGF23 with a sensitivity of 3.5 pg/mL and intra- and inter-assay coefficients of 9.5% and 6.85%, respectively. Serum Klotho was measured using the human Klotho ELISA kit (Immuno-Biological Laboratories, Takasaki, Japan), with a sensitivity of 6.15 pg/mL and intra- and inter-assay coefficients of variation of 3.1 % and 6.9 %, respectively. High-sensitivity serum C-reactive protein (hsCRP) was measured by a high-sensitivity particle enhanced immunoturbidimetric fully automated assay (Roche Diagnostics GmbH, Mannheim, Germany) in a Cobas 6000 analyzer from the same manufacturer with a sensitivity of 0.3 mg/L and intra- and inter-assay coefficients of variation of 1.6 and 8.4, respectively. Levels of the inflammatory cytokines TNFα, IL6 and IL10 were also measured by high-sensitivity ELISA methods (Quantikine®, R&D Systems, Abingdon, UK). Minimum detectable concentrations were 0.10 pg/mL, 0.70 pg/mL, and 0.50 pg/mL, respectively. Intra- and inter-assay coefficients of variability were <10.8%.

### 2.3. Immunohistochemistry

Sections of blood vessels were fixed in 4% buffered formalin for 24 h and subsequently dehydrated in ascending concentrations of ethanol, cleared in xylene, and embedded in paraffin. Blocks were trimmed and 3 µm sections were processed for immunohistochemistry. Primary antibodies used were: mouse monoclonal anti-FGFR1, 1:400 dilution (Abcam, Cambridge, UK); rabbit polyclonal anti-FGFR3, 1:300 dilution (Abcam); rabbit polyclonal anti-Klotho, 1:100 dilution (Abcam); mouse monoclonal anti-IL6, 1:300 dilution (Santa Cruz Biotechnology Inc., Dallas, TX, USA); and rabbit monoclonal anti-TNFα, 1:100 dilution (Santa Cruz Biotechnology Inc.). For the quantification analysis, a total of five images of each slide that include intima and media layers were captured and processed with a high-resolution video camera (Sony, DF-W-X710, Kōnan, Japan) connected to a light microscope (Nikon Eclipse 50i). The areas of tissue stained by the antibodies were quantified by using ImageJ software (Rasband, W.S., ImageJ, National Institutes of Health, Bethesda, MD, USA). Results are expressed in square microns.

### 2.4. Quantitative Real-Time PCR

After surgery, vascular tissue fragments were immediately placed in RNAlater® solution (Ambion (Europe) Limited, Cambridge, UK) and stored at 4 °C for subsequent RNA extraction. Total RNA was isolated by homogenization in TRI Reagent® (Sigma-Aldrich, Saint Louis, MO, USA) employing TissueRuptor (QIAGEN, Hilden, Germany). Further purification was performed using RNeasyMini kit (QIAGEN), according to manufacturer’s specifications, and stored at −80 °C. RNA was quantified using a Thermo Scientific NanoDrop Lite Spectrophotometer (Thermo Scientific, Waltham, MA, USA) and its quality was assessed by the A260/A280 ratio measured in this equipment. cDNA was obtained using a High Capacity RNA-to-cDNA kit (Applied Biosystems, Foster City, CA, USA) for further use in quantitative RT-PCR (qRTPCR). Transcripts encoding for *KL, ADAM17, TNF, IL6, IL10,* and *glyceraldehyde 3-phosphate dehydrogenase* (*GAPDH*) were measured by TaqMan quantitative PCR with TaqMan Fast Universal PCR Master Mix (Applied Biosystems, Foster City, CA, USA). TaqMan gene expression assays for each transcript (Hs00183100_m1 (KLOTHO), Hs01041915_m1 (ADAM17), Hs00174128_m1 (TNFα), Hs00985639_ml (IL6), Hs0961622_m1 (IL10), and Hs99999905_m1 (GAPDH)) were analyzed in a 7500 Fast Real-Time PCR System (Applied Biosystems). The level of target mRNA was estimated by relative quantification using the comparative method (2^−ΔΔCt^) by normalizing to *GAPDH* expression. mRNA levels were expressed as arbitrary units (a.u.). Quantification of each cDNA sample was tested in triplicate.

### 2.5. Statistical Analysis

All analyses were performed using SPSS software version 25 (IBM Corp. Armonk, NY, USA). Continuous variables are reported as mean ± SD or median with interquartile range (IQR), and categorical data as number and percent frequency. Continuous variables were checked for the normal distribution assumption using the Kolmogorov–Smirnov statistics, and those that did not satisfy the criteria were log-transformed. Comparisons between groups were performed by Chi-square test, Mann–Whitney U test, or Kruskal-Wallis test as appropriate. The Spearman correlation coefficient was calculated to assess the relationship between variables. Partial correlation analysis was performed to measure the association of serum and vascular Klotho expression with inflammatory parameters whilst controlling for the effect of covariates that were selected based on clinical relevance and the results of comparison and correlation analysis. A multiple logistic regression was performed to assess independent predictors of the presence of DFS. For this purpose, we adopted three models: In model 1, we introduced age, sex, uric acid, eGFR, and hsCRP. In model 2, we additionally included serum Klotho, FGF23, IL6, and IL10. Finally, in model 3 we adjusted the analysis for the immunoreactivity and gene expression levels of *KL*. Regarding gene expression analysis, quantification of each sample was tested in triplicate and data are expressed as arbitrary units. A two tailed *p*-value < 0.05 was considered statistically significant.

## 3. Results

### 3.1. Characteristics of the Patients and Biochemical Parameters

Eighty-five diabetic patients with established vascular disease that underwent elective revascularization surgical or lower limb amputation were initially evaluated, with 28 subjects being excluded due to exclusion criteria. Finally, 57 patients (46 males and 11 females) with a mean age of 69.8 ± 9.7 were included in the study: 20 patients were subjected to elective limb amputation by DFS, and the remaining 37 underwent elective shunt vascular surgery in the lower extremities. The specific indications for amputation included unsuccessful previous revascularization, extensive non-healing ulcers or non-healing wounds, non-reconstructable disease with persistent tissue loss, unrelenting rest pain due to muscle ischemia and gangrene. At the time of amputation, no patient had foot infection. In all cases a standard below knee transtibial amputation was performed. The clinical, biochemical, and demographic characteristics of the 57 patients included in the study are presented in Table 1.

There were no differences in any demographic characteristic between the group of patients with DFS and the remaining patients included in the study. Percentages of subjects with smoking habits, alcoholism, and hypertension were also similar in both groups. Similarly, no differences were observed in general laboratory measurements including total cholesterol, high-density lipoprotein (HDL), low-density lipoprotein (LDL), triglycerides, HbA1c, glucose, eGFR, albumin, calcium, phosphorous, and alkaline phosphatase. However, the levels of pro-inflammatory markers were higher in patients with DFS, although only reached statistical significance for hsCRP (6 (2.3–10.5) vs. 5.3 (2.4–13.6) pg/mL, *p* < 0.05) and IL6 (22 (51–28.6) vs. 6.42 (0.7–26.7) pg/mL, *p* < 0.05). Conversely, the level of anti-inflammatory cytokine IL10 was diminished in the DFS group (1.1 (0.6–6.7) vs. 3.9 (0.5–6.8) pg/mL, *p* < 0.01). Interestingly, the serum levels of Klotho were significantly diminished in patients with DFS (397.4 (318–515) vs. 617 (484–779) pg/mL, *p* < 0.01), whereas the FGF23 concentration was significantly higher (23.8 (17–32.2) vs. 15.5 (10.1–24.5) pg/mL, *p* < 0.01). On comparison with the respective levels in patients without DFS, these differences represent a median percent decrease of 37.4% for Klotho and 30.3% increase for iFGF23 concentrations in patients with DFS (Table 1 and Figure 1).

### 3.2. Immunohistochemical Analysis

For immunohistochemical determinations, serial sections of femoral or carotid were obtained from the participants. All the vascular samples obtained from the 20 patients with DFS were excised from the femoral artery. Fragments obtained from the group of 37 patients without DFS included carotid and femoral samples (21 and 16, respectively). All the fragments from the donor group were obtained from the femoral artery.

The results obtained show the presence of Klotho protein, as well as the receptors for FGF23 (FGFR1 and FGFR3), in all the samples studied (Figure 2A). Mean immunoreactivity levels for Klotho were significantly decreased in vascular sections obtained from patients with DFS as compared to sections from diabetic patients without DFS (*p* < 0.05) and from the control group (*p* < 0.01) (Figure 2B). No differences were observed in Klotho levels between patients without DFS and the control group. The difference in Klotho values between the patients with and without DFS remained when only the fragments of femoral arteries were considered in the analysis (*p* < 0.05). No differences were observed in the immunoreactivity levels for FGFR1 and FGFR3 between the two groups of diabetic patients. However, the levels of both proteins were higher in diabetic patients than in controls (*p* < 0.01). Finally, mean immunoreactivity levels of the pro-inflammatory cytokines IL6 and TNFα were higher in the group of diabetic patients than in controls (*p* < 0.05), with no differences regarding the presence or absence of DFS (Figure 2B).

### 3.3. Vascular Gene Expression

Gene expression levels of *KL, ADAM17*, and the cytokines *IL6, IL10,* and *TNF* were analyzed in all the vascular samples. Results showed that expression levels of *Klotho* were significantly reduced in DFS patients (2.04 (0.79–3.36)) when comparing to either diabetic patients without DFS (4.21 (1.98–6.61)) or to the control group (6.81 (8.51–5.12)); *p* < 0.01, for both comparisons (Figure 3). These differences represent a mean 51.4% and 70% lower expression of *KL* in the vascular wall of DFS patients compared to patients without DFS and to control individuals, respectively. *ADAM17* expression levels were similar among all the groups. Concerning inflammatory cytokines, the vascular gene expression levels of *TNF* and *IL6* were higher in the group of patients with DFS when compared to control subjects (*p* < 0.05 and *p* < 0.01, respectively), although only the *IL6* levels differed between patients with and without DFS (2.04 (0.79–3.36) vs. 1.41 (0.34–7.83), *p* < 0.01). This difference represented a mean 30.9% higher expression of *IL6* in the vascular wall of patients with DFS. Finally, the expression levels of the anti-inflammatory cytokine *IL10* in both groups of diabetic patients were reduced with respect to the control subjects (*p* < 0.05).

### 3.4. Correlations and Multivariate Analysis

A correlation analysis was performed between Klotho levels and the other parameters determined in the study (Table 2). A statistically significant positive correlation was found between eGFR and serum and vascular expression levels of Klotho (*r* = 0.329 and *r* = 0.354, respectively, *p* < 0.01). Vascular Klotho levels were also directly related to serum Klotho concentrations (*r* = 0.288, *p* < 0.05) and inversely related to serum uric acid (*r* = 0.342, *p* < 0.01), serum creatinine (*r* = −0.389, *p* < 0.01), and vascular protein levels of IL6 (*r* = −0.337, *p* < 0.01). Finally, vascular expression levels of *KL and ADAM17* showed a positive correlation (*r* = 0.369, *p* < 0.01).

Partial correlation analysis was performed to measure the associations of serum and vascular Klotho expression with inflammatory parameters after controlling for covariates (age, eGFR, HbA1c, hemoglobin, hematocrit, HDL, LDL, triglycerides, and uric acid) (Table 3). In the first analysis, soluble Klotho was significantly related to serum CRP (*r* = −0.30, *p* < 0.05), with a marginal association with serum IL6. In the second analysis, vascular mRNA expression of *KL* was directly associated with the expression of *ADAM17* (*r* = 0.41, *p* = 0.01), whilst a significant inverse association was observed with *IL6* expression (*r* = −0.31, *p* < 0.05). Regarding vascular immunoreactivity for Klotho, this parameter showed an inverse association with vascular immunoreactivity for IL6 (*r* = −0.29, *p* < 0.04).

Finally, the multivariate logistic regression analysis using the presence/absence of DFS as the dependent variable showed that higher serum Klotho concentrations and gene expression levels in the vascular wall were associated with a lower risk for DFS, with immunoreactivity values for vascular Klotho showing a marginal association. On the contrary, higher serum FGF23 was related to a higher risk for this complication (Table 4).

## 4. Discussion

The main results of the present study show that patients with DFS have lower serum concentrations of soluble Klotho and elevated concentrations of FGF23, while at the vascular level, both immunoreactivity and gene expression levels of *KL* were also diminished. Moreover, these alterations in the components of the FGF23/Klotho system are independently associated with the presence of DFS. These findings have been not reported previously and suggest an intriguing possibility about the potential role of FGF23/Klotho in the development and/or progression of DFS.

Vascular disease is a critical factor underlying the pathogenesis of DFS [22]. The FGF23/Klotho system has been decisively related to various processes associated with vascular damage. Recent works suggest that FGF23 can exert direct effects on the cardiovascular system. It has been proven that at high concentrations, FGF23 is able to establish low affinity bonds with its receptors, independently of the presence of Klotho, which could cause deleterious effects on multiple organs and tissues [14]. From a clinical point of view, elevated levels of FGF23 have been associated with the presence of vascular dysfunction [15], as well as with a greater severity of atherosclerotic vascular damage [23]. A direct effect of FGF23 on the integrity of vascular tissue has been proposed, a hypothesis that has been reinforced by the recent demonstration of the expression of its receptors, as well as Klotho, in the human vascular wall [16,18]. Our results show that patients with DFS present not only higher serum levels of FGF23, but also increased vascular expression of FGFRs. These findings point out the possibility that high FGF23 could exert direct pathogenic actions on the vasculature even though vascular levels of Klotho are diminished.

On the other hand, Klotho deficiency has been also associated with vascular damage. In murine models, absence of Klotho causes a syndrome of accelerated aging, which included arteriosclerosis and vascular calcifications [24], alterations that could be reversed by administration of the *Kl* gene or by parabiosis with the wild mice [25]. More recent studies have confirmed the protective effects of Klotho on the vascular system, including its participation in the maintenance of endothelial homeostasis and vascular functionality [26]. In the clinical setting, low serum Klotho concentrations in adult subjects without known risk factors for cardiovascular disease have been related to a reduced capacity of flow-mediated dilation of the brachial artery and higher values of epicardial fat thickness and carotid artery intima-media thickness, suggesting that Klotho deficiency may be an early predictor of subclinical atherosclerosis [27]. Moreover, recent works have related the presence of established clinical atherosclerotic disease with low serum Klotho levels [19,21,28]. To the best of our knowledge, only one previous study has evaluated the serum Klotho concentrations in T2DM in relation to diabetic complications. In that study, Zhang et al. [29] studied 102 T2DM patients with DFS and observed that these individuals presented a 44.79% reduction in serum Klotho as compared with non-diabetic controls, with no differences with respect to patients with other complications but no diabetic foot. In the present work, patients with DFS presented a median percent decrease of 37.4% in serum Klotho concentrations as compared with diabetic subjects without DFS. These differences could be explained by the distinct subjects’ characteristics between the studies, since in the work by Zhang et al. the patients were younger (mean age, 56.1 vs. 71.8 years), with a greater female proportion (43.1% vs. 25%), lower mean BMI (27.4 vs. 30.8 kg/m^2^), and more importantly, all of our patients had severe DFS with lower limb amputation, whereas in the study by Zhang et al. the criteria for the DFS group included foot ulcers, deformity, infection, or gangrene. Therefore, it is possible to speculate that reduction in serum Klotho may reflect a progressive process that is associated with the severity of DFS.

Importantly, the endogenous expression of *KL* in human arteries has been recently demonstrated, an issue of special relevance since the vascular wall may constitute a source of soluble Klotho, with putative autocrine and/or paracrine protective beneficial effects on vascular homeostasis [16,18]. Clinical data regarding the relationship between vascular *KL* expression and atherosclerotic disease are scarce. In previous works, our group reported that low vascular *KL* gene expression was associated with the presence and severity of coronary artery disease [21] and clinical atherosclerotic disease [19] independently of established cardiovascular risk factors. The present study reports for the first-time data about vascular expression of Klotho, both at gene and protein levels, in patients with DFS. In particular, we have detected a 51.4% lower gene expression and a 52.1% lower protein expression of Klotho in vascular samples of patients with DFS as compared with diabetic subjects without this complication. Overall, these findings may be relevant from a mechanistic perspective since Klotho has been involved in maintaining the health of the vasculature, and thus, disruption of Klotho homeostasis may be an important factor in the development and progression of atherosclerotic disease in general, and DFS in particular.

Finally, it is important to note the interplay between inflammation and Klotho, since inflammation induces the reduction of *KL* expression, both in the kidney as well as in the vessels [16,20,30]. Our results show that subjects with DFS presented higher serum concentrations of hsCRP and IL6, and reduced levels of the anti-inflammatory cytokine IL10. In addition, the immunoreactivity levels of TNFα and IL6 were increased in the two groups of diabetic patients as compared with non-diabetic subjects. Regarding gene expression, diabetic subjects presented a higher expression of *TNF* and lower levels of *IL10* compared to control subjects, whereas IL6 expression was significantly higher in patients with DFS as compared with non-diabetic individuals and diabetic subjects without DFS. Importantly, after adjusting for the effect of other variables, arterial *KL* mRNA levels showed an independent negative relationship with *IL6* expression, while vascular Klotho immunoreactivity was also negatively associated with immunoreactivity levels for IL6, supporting the role of inflammation as an inducer of vascular Klotho downregulation in DFS.

Although this study provides novel information about the relationship between DFS and FGF23/Klotho, several limitations need to be acknowledged. First, the small sample size since this work was designed as a proof-of-concept study, which might imply that some of our results did not reach statistical significance and limit the ability to adequately describe the closeness of the relationship between the variables; however, the results indicate that the initial assumptions can be valid for generating new hypotheses. Second, given the cross-sectional nature of the study, we can only demonstrate associations without definitive inferences on their direction or causality. Third, serum levels of other factors related to Klotho and FGF23, such as vitamin D, which may potentially impact DFS, were not measured, and therefore, their potential influence on the results may not be completely ruled out.

## 5. Conclusions

In conclusion, this paper describes that patients with DFS present lower serum concentrations of Klotho and elevated FGF23, and reports the reduced expression of Klotho, both at the protein and gene level, in their vascular beds. Importantly, these parameters were independently associated with DFS. Overall, these findings point to a new pathophysiological pathway potentially involved in DFS. Further studies are imperative to elucidate the role of the FGF23/Klotho system in the development and progression of this complication.

## Figures and Tables

**Figure 1 jcm-08-00448-f001:**
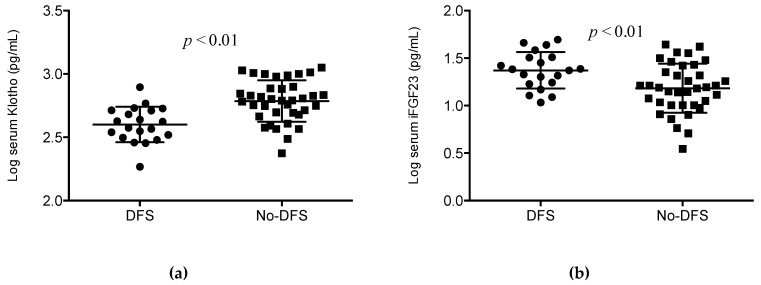
Differences in the log-transformed blood levels of (**a**) Klotho and (**b**) FGF23 between patients with and without diabetic foot syndrome (DFS).

**Figure 2 jcm-08-00448-f002:**
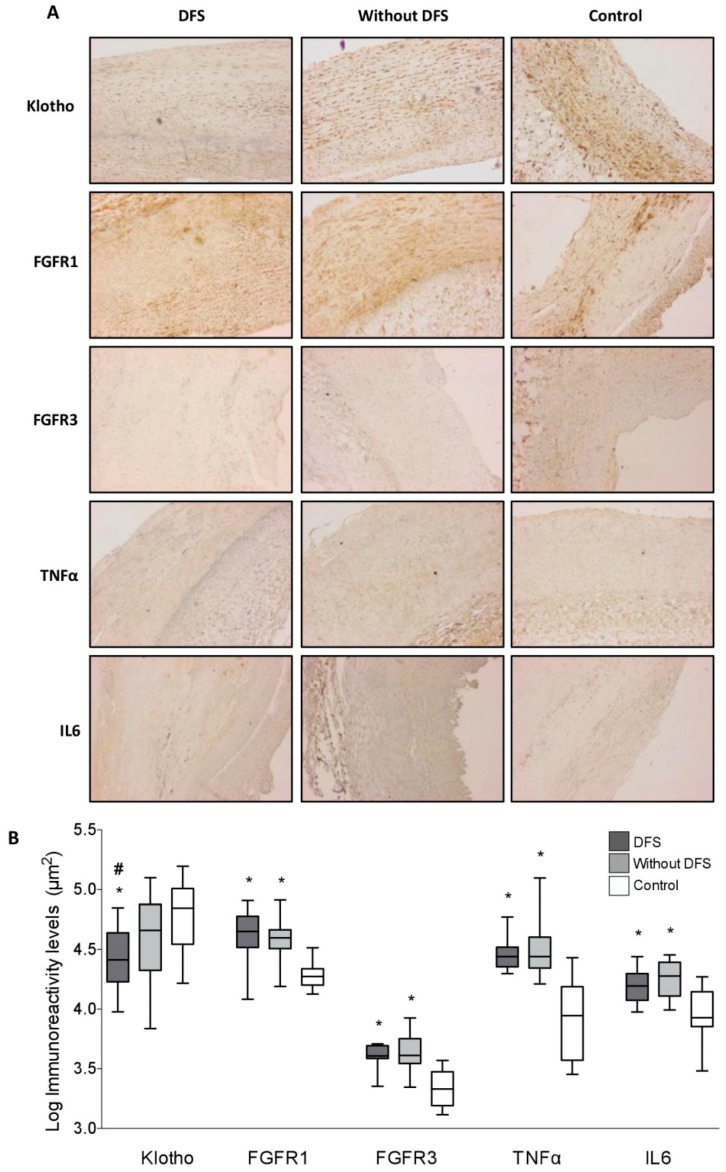
(**a**) Immunohistochemical staining for Klotho, FGFR1 (fibroblast growth factor receptor (1), FGFR3, TNFα (tumor necrosis factor α), and IL6 (interleukin 6) in arterial sections of patients with and without diabetic foot syndrome (DFS), and control donors (magnification 4×). (**b**) Mean immunoreactivity levels. * *p* < 0.01 vs. control group; # *p* < 0.05 vs. patients without DFS.

**Figure 3 jcm-08-00448-f003:**
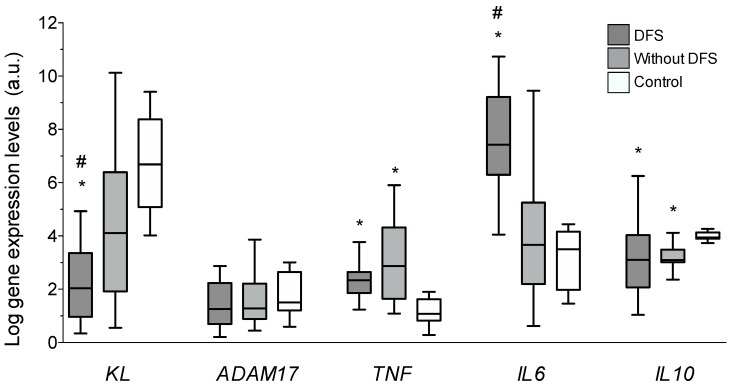
Gene expression levels of *KL*, *ADAM17* (A Disintegrin and A Metalloproteinase 17), *TNF* (tumor necrosis factor α), *IL6* (interleukin 6), and *IL10* determined by quantitative RT-PCR in vascular walls of patients with diabetic foot syndrome (DFS), without DFS, and controls. * *p* < 0.01 vs. control group; # *p* < 0.05 vs. patients without DFS.

**Table 1 jcm-08-00448-t001:** Clinical characteristics and biochemical assessments of the diabetic patients included in the study.

Variable	Overall	DFS	No. DFS	*p*–Value
*n*	57	20	37	
Age (years)	69.8 ± 9.7	71.8 ± 10.2	68.7 ± 9.4	NS
Male gender, *n* (%)	46 (80)	15 (75)	31 (84)	NS
BMI (kg/m^2^)	30.1 ± 3.8	30.8 ± 4.7	29.7 ± 3.3	NS
Hypertension, *n* (%)	51 (89.4)	17 (85)	34 (91.9)	NS
Alcoholism, *n* (%)	22 (38.6)	7 (35)	15 (40.5)	NS
Smoking habits, *n* (%)	36 (63.2)	12 (60)	24 (64.8)	NS
Total Cholesterol (mg/dL)	159.2 ± 38.9	159 ± 45.8	159.3 ± 35.3	NS
HDL (mg/dL)	42.4 ± 10.2	40 ± 9.8	43.7 ± 10.4	NS
LDL (mg/dL)	84.8 ± 29.4	87.5 ± 33.4	83.3 ± 27.3	NS
Triglycerides, mg/dL	155 ± 80.2	156.6 ± 100	154.2 ± 68.4	NS
HbA1c, %	7.3 ± 1.4	7.3 ± 1.9	7.2 ± 1.3	NS
Glucose	132.9 ± 42.8	126.3 ± 40.1	136.6 ± 44.4	NS
eGFR, mL/min/1.73 m^2^	77.5 ± 24.7	76.6 ± 24	77.8 ± 25.4	NS
Albumin (g/L)	3.8 ± 0.7	3.7 ± 0.8	3.8 ± 0.6	NS
Calcium (mg/dl)	9.1 ± 0.6	9 ± 0.6	9.1 ± 0.6	NS
Phosphorous (mg/dl)	3.6 ± 0.5	3.6 ± 0.5	3.6 ± 0.5	NS
AP (mU/mL)	78.8 ± 58.1	77.9 ± 43.3	79.2 ± 65.3	NS
hsCRP (mg/dL)	5.8 (2.4–11.9)	6 (2.3–10.5)	5.3 (2.4–13.6)	<0.05
IL6, pg/mL	11.2 (1.1–28.1)	22 (5.1–28.6)	6.42 (0.7–26.7)	<0.05
IL10, pg/mL	1.3 (0.5–6.7)	1.1 (0.6–6.7)	3.9 (0.5–6.8)	<0.01
TNFα, pg/mL	1.3 (0.8–1.3)	1.4 (0.9–2.1)	1.1 (0.7–1.7)	NS
KLOTHO, pg/mL	532.8 (375–677)	397.4 (318–515)	617.3 (484–779)	<0.01
iFGF23, pg/mL	17.5 (12–27)	23.8 (17–32)	15.5 (10.1–24.5)	<0.01

DFS, diabetic foot syndrome; BP, blood pressure; BMI, body mass index; HDL, high-density lipoprotein; LDL, low-density lipoprotein; HbA1c, hemoglobin A1c; eGFR, estimated glomerular filtration rate; AP, alkaline phosphatase; hsCRP, high-sensitivity C-reactive protein; IL, interleukin; TNF, tumor necrosis factor; FGF23, fibroblast growth factor 23; NS, not significant. *p*-value reflects differences between with DFS and without DFS.

**Table 2 jcm-08-00448-t002:** Correlations of serum and vascular gene expression levels of Klotho in diabetic patients.

	Log Serum KL		Log Vascular *KL*	
Variable	*r*	*p*-Value	*r*	*p*-Value
BMI (kg/m^2^)	0.129	NS	−0.062	NS
Total Cholesterol (mg/dL)	0.055	NS	−0.139	NS
HDL (mg/dL)	0.255	<0.05	−0.131	NS
LDL (mg/dL)	0.021	NS	−0.152	NS
Triglycerides, mg/dL	−0.058	NS	0.134	NS
HbA1c, %	−0.034	NS	0.131	NS
Hemoglobin	0.344	<0.05	0.104	NS
Albumin (g/L)	−0.038	NS	0.065	NS
Hematocrit	0.341	<0.05	0.109	NS
Calcium (mg/dl)	0.206	NS	−0.51	NS
Phosphorous (mg/dl)	0.071	NS	0.17	NS
AP (mU/mL)	−0.62	NS	0.03	NS
Uric acid (mg/dL)	0.096	NS	−0.342	<0.01
eGFR, mL/min/1.73 m^2^	0.329	<0.01	0.354	<0.01
hsCRP (mg/dL)	−0.196	<0.05	0.076	NS
Serum IL6 (pg/mL)	−0.204	NS	−0.094	NS
Serum IL10 (pg/mL)	0.172	NS	−0.099	NS
Serum TNFα (pg/mL)	−0.159	NS	−0.114	NS
Serum Klotho (pg/mL)			0.125	NS
Serum iFGF23 (pg/mL)	0.042	NS	−0.064	NS
VIR Klotho (µm^2^)	0.288	<0.05	0.097	NS
VIR FGFR1 (µm^2^)	−0.15	NS	0.158	NS
VIR FGFR3 (µm^2^)	0.153	NS	−0.106	NS
VIR IL6 (µm^2^)	−0.124	NS	−0.337	<0.01
VIR TNFα (µm^2^)	0.076	NS	−0.177	NS
Log *KL* mRNA (a.u.)	0.125	NS		
Log *ADAM17* mRNA (a.u.)	0.183	NS	0.369	<0.01
Log *IL6* mRNA (a.u.)	−0.81	NS	0.133	NS
Log *IL10* mRNA (a.u.)	−0.069	NS	0.223	NS
Log *TNF* mRNA (a.u.)	0.069	NS	0.183	NS

BMI, body mass index; HDL, high-density lipoprotein; LDL, low-density lipoprotein; HbA1c, hemoglobin A1c; AP, alkaline phosphatase; eGFR, estimated glomerular filtration rate; hsCRP, high-sensitivity C-reactive protein; IL, interleukin; TNF, tumor necrosis factor; KL, Klotho; a.u., arbitrary units; FGF23, fibroblast growth factor 23; VIR, vascular immunoreactivity; NS, not significant.

**Table 3 jcm-08-00448-t003:** Association between serum and vascular expression levels of Klotho with inflammatory parameters assessed by partial correlation analysis.

**Serum Klotho Concentrations**		
Serum hsCRP	*r* = −0.30	*p* < 0.05
Serum IL6	*r* = −0.23	*p* = 0.10
Serum IL10	*r* = 0.20	*p* = 0.17
Serum TNFα	*r* = −0.12	*p* = 0.39
**Vascular mRNA *KL* Expression Levels**		
Expression *ADAM17*	*r* = 0.41	*p* = 0.001
Expression *IL6*	*r* = −0.31	*p* < 0.05
Expression *IL10*	*r* = 0.19	*p* = 0.20
Expression *TNF*	*r* = 0.03	*p* = 0.81
**Vascular Klotho Immunoreactivity Levels**		
Immunoreactivity IL6	*r* = −0.29	*p* < 0.05
Immunoreactivity TNFα	*r* = −0.06	*p* = 0.64

hsCRP, high-sensitivity C-reactive protein; IL6, interleukin 6; IL10, interleukin 10; TNFα, tumor necrosis factor α; ADAM17, A Disintegrin and A Metalloproteinase 17. Covariates: age, estimated glomerular filtration rate, glycated hemoglobin, hemoglobin, hematocrit, HDL, LDL, triglycerides, and uric acid.

**Table 4 jcm-08-00448-t004:** Multivariate logistic regression analysis for the presence of DFS.

***Model 1***		
**Independent Variable**	**Odds Ratio (95% CI)**	***p*-Value**
Age	1.03 (0.97 to 1.10)	0.28
Gender	0.59 (0.13 to 2.78)	0.51
Uric acid	0.96 (0.64 to 1.45)	0.87
eGFR	1.05 (0.97 to 1.03)	0.72
hsCRP	0.99 (0.93 to 1.06)	0.95
***Model 2 (Model 1 + Serum Klotho, FGF23, IL6 and IL10)***		
**Independent Variable**	**Odds Ratio (95% CI)**	***p*-Value**
Age	1.00 (0.91 to 1.10)	0.82
Gender	0.75 (0.08 to 7.04)	0.80
Uric acid	1.09 (0.57 to 2.08)	0.78
eGFR	1.01 (0.97 to 1.05)	0.49
hsCRP	0.97 (0.89 to 1.07)	0.64
Serum Klotho	0.99 (0.98 to 0.99)	<0.01
Serum FGF23	1.10 (1.05 to 1.21)	<0.05
Serum IL6	1.00 (0.98 to 1.01)	0.88
Serum IL10	1.06 (0.85 to 1.32)	0.60
***Model 3 (Model 2 + Soluble Klotho)***		
**Independent Variable**	**Odds Ratio (95% CI)**	***p*-Value**
Age	0.91 (0.78 to 1.06)	0.23
Gender	4.98 (0.17 to 14.63)	0.35
Uric acid	0.64 (0.27 to 1.50)	0.31
eGFR	1.04 (0.98 to 1.12)	0.15
hsCRP	1.16 (0.93 to 1.33)	0.23
Serum Klotho	0.99 (0.98 to 0.99)	<0.05
Serum FGF23	1.22 (1.02 to 1.47)	<0.05
Serum IL6	0.99 (0.97 to 1.01)	0.53
Serum IL10	0.97 (0.77 to 1.23)	0.83
VIR Klotho	1.00 (0.99 to 1.00)	0.06
KL gene expression	0.66 (0.43 to 0.99)	<0.05

eGFR, estimated glomerular filtration rate; hsCRP, high-sensitivity C-reactive protein; FGF23, fibroblast growth factor 23; IL, interleukin; VIR, vascular immunoreactivity.

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
