# Peer review of "FGF23 and Klotho Levels are Independently Associated with Diabetic Foot Syndrome in Type 2 Diabetes Mellitus"

_jcm, 2019, doi:10.3390/jcm8040448_

Reviewer 1 Report

In this manuscript, the authors have examined the expression level of FGF23 and Klotho are independently associated with diabetic foot syndrome in type 2 diabetes mellitus. The study plan is very good and innovative. It’s a very well written manuscript. Study design, results and presentation of data is really good. The way authors have written the manuscript is remarkable specifically describing the shortcomings of their study design, which is very common with patient data (clinical cases). The concept is novel and results are very intriguing. It will be interesting to see how far these findings can go. As results looks very convincing with patient data presenting Klotho expression significantly downregulated in DFS patient. I have no major or minor comments to mention.

Author Response

We thank the reviewer for the kind comments. We agree that our results open a new field of research focused on the effects of the variations of Klotho in the DFS.

Reviewer 2 Report

 Points to clarify

Line 171

“Eighty-five diabetic patients with established vascular disease that underwent elective vascular  surgery were initially evaluated”

Did the DFS patients undergo vascular surgery and amputation?

 Line 173

“Within this group, 20 patients were subjected to limb amputation by DFS, and the remaining 37 underwent another type of elective surgery including endarterectomy or shunt vascular surgery in the lower  extremities.”

 Diabetic Foot Syndrome is a very wide embracing term

Were the 20 DFDs patients complicated by foot infection which could have affected the results?

Why were they having an amputation?

What was the type of amputation?

Author Response

We thank the reviewer’s comments. We have modified the manuscript according to his suggestions. These modifications have been included in the first paragraph of the Results section (lines 171-179).

1. Did the DFS patients undergo vascular surgery and amputation?

Sorry so much. We recognize that this part was not clear enough and now we have modified it in the manuscript to clarify that the 85 patients initially considered had established vascular disease and were underwent an elective vascular surgery procedure or an elective lower limb amputation. According to the exclusion criteria 28 patients were excluded, and therefore, 57 patients were included in the study: 37 subjects under a surgical revascularization procedure and 20 under an elective lower limb amputation.

 2. Line 173 “Within this group, 20 patients were subjected to limb amputation by DFS, and the remaining 37 underwent another type of elective surgery including endarterectomy or shunt vascular surgery in the lower  extremities.” Diabetic Foot Syndrome is a very wide embracing term.

We agree with the reviewer’s comment. The diabetic foot syndrome is defined as as ulceration, destruction and/or infection of deep tissues associated with neurological abnormalities and various degrees of peripheral vascular disease*; therefore, as indicated by this definition, one or more of these conditions may coincide. In our study, the patients with DFS presented severe peripheral vascular disease and harsh complications with no therapeutic options. 

* International Consensus on the Diabetic Foot: Guidance on the Diabetic Foot. International Working Group on the Diabetic Foot 2015. The Hague, Netherlands. (USB) www.iwgdf.org

3.  Were the 20 DFDs patients complicated by foot infection which could have affected the results?

Amputations in the 20 patients with DFS were elective due severe peripheral artery disease with harsh complications. At the time of amputation none these patients was complicated by foot infection.

4. Why were they having an amputation?

The indications of amputation in the patients with DFS in our study included severe peripheral vascular disease with harsh complications: unsuccessful previous revascularization, non-healing ulcers or non-healing wounds, nonreconstructable disease with persistent tissue loss, unrelenting rest pain due to muscle ischemia and gangrene.

5. What was the type of amputation?

In all cases a standard below knee transtibial amputation was performed.